# Uncovering the Localization and Function of a Novel Read-Through Transcript ‘*TOMM40-APOE*’

**DOI:** 10.3390/cells13010069

**Published:** 2023-12-28

**Authors:** Shichen Chang, Satoru Torii, Jun Inamo, Kinya Ishikawa, Yuta Kochi, Shigeomi Shimizu

**Affiliations:** 1Department of Pathological Cell Biology, Medical Research Institute, Tokyo Medical and Dental University (TMDU), 1-5-45 Yushima, Bunkyo-ku, Tokyo 113-8510, Japantoripcb@tmd.ac.jp (S.T.); 2Department of Personalized Genomic Medicine for Health, Tokyo Medical and Dental University (TMDU), 1-5-45 Yushima, Bunkyo-ku, Tokyo 113-8510, Japan; 3Department of Genomic Function and Diversity, Medical Research Institute, Tokyo Medical and Dental University (TMDU), 1-5-45 Yushima, Bunkyo-ku, Tokyo 113-8510, Japan

**Keywords:** read-through transcript, gene fusion, TOMM40, APOE, mitochondria, cell death

## Abstract

Recent advancements in genome analysis technology have revealed the presence of read-through transcripts in which transcription continues by skipping the polyA signal. We here identified and characterized a new read-through transcript, *TOMM40-APOE*. With cDNA amplification from THP-1 cells, the *TOMM40-APOE3* product was successfully generated. We also generated *TOMM40-APOE4*, another isoform, by introducing point mutations. Notably, while APOE3 and APOE4 exhibited extracellular secretion, both TOMM40-APOE3 and TOMM40-APOE4 were localized exclusively to the mitochondria. But functionally, they did not affect mitochondrial membrane potential. Cell death induction studies illustrated increased cell death with TOMM40-APOE3 and TOMM40-APOE4, and we did not find any difference in cellular function between the two isoforms. These findings indicated that the new mitochondrial protein TOMM40-APOE has cell toxic ability.

## 1. Introduction

In recent years, genome analysis technology has made remarkable progress, enabling long-read sequencing [1,2,3]. As a result, many unannotated transcripts and isoforms have been found [4]. Among them, there are read-through transcripts in which transcription continues by skipping the polyA signal, which originally serves as a stop codon, and fusion mRNA containing downstream gene sequences is produced [5]. One such read-through transcript is a fusion transcript of *TOMM40*, a gene encoding the mitochondrial outer membrane protein TOMM40, and *APOE*, encoding the secreted protein APOE. They are neighboring genes and are considered risk genes for Alzheimer’s disease [4,6,7] because their SNPs are strongly correlated with the accumulation and aggregation of amyloid-β and the disease severity of Alzheimer’s disease [8,9,10,11]. Therefore, the read-through transcript of *TOMM40-APOE* may also influence the pathogenesis of Alzheimer’s disease.

TOMM40 localizes to the mitochondrial outer membrane as one of the members of the Tim/Tom protein complex that transports mitochondrial proteins from the cytosol to mitochondria [12]. APOE is a member of a family of lipid-binding proteins called apolipoproteins, which are mainly found in blood and are responsible for transporting cholesterol, etc. [13,14]. There are three isoforms of APOE that differ in amino acids 112 and 158, with APOE3 (Cys112, Arg158) being the basic one, while APOE4 (Arg112, Arg158) is known to increase the incidence of Alzheimer’s disease [8,15].

Although the existence of the *TOMM40-APOE* transcript is known [4], the nature of the protein and how it is involved in cellular phenomena have not been analyzed at the protein level, which we clarified in this study. In addition, since there are *APOE* alleles, the normal types *E3* and *E4*, with a higher risk of Alzheimer’s disease, we analyzed the differences between *TOMM40-APOE3* and *TOMM40-APOE4*.

## 2. Materials and Methods

### 2.1. Construction of cDNA Constructs

DNA fragments were PCR-amplified from a human monocytic cell line (THP-1 cells) cDNA using primer 1 (CACAGGCGACCATGGGGAACGTGTTG, *TOMM40* N-terminal sequence) and primer 2 (GCTTCGGCGTTCAGTGATTG, *APOE* C-terminal sequence), followed by a second PCR to introduce *EcoRI* and *XhoI* restriction enzyme sites using PrimeSTAR GXL polymerase (Takara Bio, Kusatsu, Japan). The resulting PCR products were ligated into the pcDNA3-HA vector. Point mutations in pcDNA3-APOE3 and pcDNA3-TOMM40-APOE3 were introduced with PCR amplification using Pfu Turbo polymerase (Agilent Technologies, Santa Clara, CA, USA). Validation of all sequences was performed using Sanger sequencing.

### 2.2. Immunocytochemistry

Neuro2a cells were transfected with one of the following vectors: APOE3-HA, APOE4-HA, TOMM40-APOE3-HA, or TOMM40-APOE4-HA. Four hours after transfection, the culture medium was replaced with a medium containing 2% fetal bovine serum (FBS) and 10 µM retinoic acid to induce neuronal differentiation. Then, 28 h after transfection, cells were fixed with 4% paraformaldehyde for 10 min and then permeabilized using 0.5% Triton X-100 (for mitochondria and endoplasmic reticulum) or 50 μg/mL Digitonin (for Golgi apparatus) for 5 min. Subsequently, cells were incubated with primary antibodies: anti-HA (Santa Cruz, Dallas, TX, USA) (dilution 1:200) and one of the following: anti-Tom20 (Santa Cruz, Dallas, TX, USA), KDEL (Enzo, Farmingdale, NY, USA), or GS28 (BD, Franklin Lakes, NJ, USA) (each at dilution 1:200) overnight at 4 °C. Following washing, cells were incubated with secondary antibodies conjugated Alexa Fluor 488 or 555 (Invitrogen, Waltham, MA, USA) (dilution 1:500) for 1 h at room temperature, mounted with Prolong Diamond Antifade (Invitrogen, Waltham, MA, USA) reagent containing DAPI, and examined with a laser-scanning confocal microscope (LSM710, Zeiss, Oberkochen, Germany).

### 2.3. Analysis of Mitochondrial Membrane Potential

Neuro2a cells were cultured in 6-well plates and co-transfected with pmax-GFP vectors in combination with one of the following vectors: APOE3-HA, APOE4-HA, TOMM40-APOE3-HA, or TOMM40-APOE4-HA. Four hours after transfection, the culture medium was replaced with a medium containing 2% FBS and 10 µM retinoic acid to induce neuronal differentiation. Then, 28 h after transfection, the culture medium was aspirated, and MitoTracker Red (Invitrogen, Waltham, MA, USA) (25 nM in MEM containing 2% FBS) was added and incubated for 15 min at 37 °C. Following a PBS wash, cells were harvested and suspended in 1 mL of PBS. Subsequently, flow cytometry analysis was conducted. Initially, cell populations were isolated from cellular debris using forward scatter (FSC) and side scatter (SSC), followed by the separation of single-cell populations using forward scatter-area (FSC-A) and forward scatter-height (FSC-H). The fluorescence of GFP, representing transfected cells, and MitoTracker Red, indicating mitochondrial membrane potential, were quantified. Neuro2a cells solely transfected with pmax-GFP vectors were used as a negative control, while cells solely transfected with pmax-GFP vectors and treated with 50 µM carbonyl cyanide *m*-chlorophenyl hydrazone (CCCP) for 15 min served as a positive control.

### 2.4. Flow Cytometry Analysis of Cell Death

Neuro2a cells were cultured in 6-well plates and co-transfected with piRFP670-N1 vectors along with one of the following vectors: APOE3-HA, APOE4-HA, TOMM40-APOE3-HA, or TOMM40-APOE4-HA. Four hours after transfection, the culture medium was substituted with medium containing 2% FBS and 10 µM retinoic acid to induce neuronal differentiation. At predefined time points (28 h, 40 h, and 52 h post-transfection), the culture medium was collected. Subsequently, cells underwent a wash with PBS, and both the wash solution and the cells were harvested. The assessment of cell death was conducted using the propidium iodide (PI) uptake assay, which was followed by flow cytometry analysis. The fluorescence signals of piRFP670, representing transfected cells, and PI, indicating cell death levels, were quantified. As a negative control, Neuro2a cells solely transfected with piRFP670-N1 vectors were used.

### 2.5. Immunoblotting

Cells were washed with ice-cold PBS, frozen, and then lysed in cell lysis buffer containing 20 mM Hepes (pH 7.5), 100 mM NaCl, 1.5 mM MgCl_2_, 1 mM EGTA, 10 mM Na_2_P_2_O_7_, 10% glycerol, 1% Nonidet P-40, 1 mM dithiothreitol, 1 mM Na_3_VO_4_, and 1% protease inhibitor cocktail (Nacalai, Kyoto, Japan). Following vortexing for 15 s, protein concentrations were determined using a Protein Assay BCA Kit (Nacalai, Kyoto, Japan). Sample buffer (5× stock, containing 0.3 M Tris (pH 6.8), 10% SDS, 50% Glycerol, 5mg Bromophenol Blue) was prepared with a 4:1 ratio of 2-mercaptoethanol, and samples were mixed with sample buffer in a 3:1 ratio. Samples were heated at 100 °C for 10 min, and the resulting supernatants were loaded onto 5–20% SDS-polyacrylamide gels. After electrophoresis, the proteins were transferred to a PVDF membrane (Millipore (Burlington, MA, USA)). The membranes were blocked with 5% skim milk in TBS-T (TBS containing 0.05% Tween-20) for 1 h at room temperature and subsequently incubated with primary antibodies (anti-HA, dilution 1:200) overnight at 4 °C. Following washing with TBS-T, the membranes were exposed to a horseradish peroxidase-labeled secondary antibody and visualized using Chemi-Lumi One Super reagent (Nacalai, Kyoto, Japan).

### 2.6. Immunoprecipitation Using Cell Media

Cell media were harvested and centrifuged at 800× *g* for 3 min to remove cell debris. Immunoprecipitation using cell media was performed using an anti-HA antibody in the presence of protein-G Sepharose (GE Healthcare, Chicago, IL, USA) for 18 h at 4 °C. The beads were then washed three times with PBS containing 1 mM dithiothreitol, 1 mM Na_3_VO_4_, and 1% protease inhibitor cocktail. The immunoprecipitated proteins were loaded onto 5–20% SDS-polyacrylamide gels. After electrophoresis, the proteins were blotted onto a PVDF membrane (Millipore (Burlington, MA, USA)). Immunoblotting was performed with an EasyBlot anti-rabbit IgG kit (GeneTex, Irvine, CA, USA) to avoid the detection of non-specific IgG bands. The membranes were blocked with 5% skim milk (EasyBlot anti-rabbit IgG kit (GeneTex, Irvine, CA, USA)) in TBS containing 0.05% Tween-20 (TBS-T) and incubated with the primary antibody (anti-HA, 561, MBL) at 4 °C overnight. After washing with TBS-T, the membranes were incubated with a horseradish peroxidase-labeled secondary antibody (EasyBlot anti-rabbit IgG kit (GeneTex, Irvine, CA, USA)) and visualized with Chemi-Lumi One Super reagent (Nacalai, Kyoto, Japan).

### 2.7. Quantification of TOMM40-APOE Transcripts in Human Brains

We analyzed short-read RNA-sequencing datasets obtained from postmortem human hippocampus brains from 8 AD and 10 non-AD subjects registered in Gene Expression Omnibus (GSE173955). From raw sequenced reads, 3′ ends with low-quality bases (Phred quality score < 20) and adaptor sequences were trimmed using Trim_Galore (version: 0.6.5). An expression matrix was generated using kallisto [16] with default parameters. To quantify transcripts, we used the de novo transcript annotation derived from long-read sequencing using 29 immune cell types [4].

### 2.8. Data Analysis

Confocal fluorescence microscopy data were analyzed utilizing Zen software 2012 (Zeiss) and ImageJ software (version: 1.54f). Flow cytometry data were processed using BD FACSDiva (version: 6.1.3) and FlowJo software (version: 10.9.0). One-way analysis of variance (ANOVA) was used for multiple comparisons, followed by post hoc Tukey tests. All statistical analyses were performed using GraphPad Prism software (version: 5.04). A *p*-value less than 0.05 was considered statistically significant.

## 3. Results

### 3.1. Purification of TOMM40-APOE Products

*TOMM40-APOE* is a read-through transcript that skips the polyA signal of *TOMM40* [4] (Figure 1A,B). We first purified this transcript from the cDNA library of THP-1 cells, a human monocytic leukemia cell line, by performing PCR, and the corresponding product was amplified (Figure 1C). This product was then incorporated into a mammalian expression vector and sequenced, which confirmed the generation of expected *TOMM40-APOE3*. The differences between this fusion transcript and the transcripts of *TOMM40* (Transcript ID: ENST00000426677.7) and *APOE* (Transcript ID: ENST00000252486.9) were as follows: (1) The cDNA sequence of *TOMM40-APOE* lacks a 92 bp sequence at the 5’ untranslated region (Figure 1A: *1). But the translation start site ATG is the same. (2) The nucleotide sequence at the ninth exon of *TOMM40* and the first exon of *APOE* is transformed into the ninth intron in the *TOMM40-APOE* read-through transcript. This leads to the loss of the 660 bp sequence of *TOMM40*, including the 140 bp protein coding sequence with the stop codon TGA, resulting in the deletion of the C-terminal 46 amino acid sequence of TOMM40 (Figure 1A: *2, Figure 1B). (3) The 23 bp sequence before the start codon ATG of the second exon of *APOE*, originally untranslated, translates into the eight amino acids (DWPITGRK) in the *TOMM40-APOE* transcript, followed by a sequence identical to *APOE* (Figure 1A: *3, Figure 1B). (4) The read-through transcripts utilize the canonical splice donor and acceptor sites at *TOMM40* Ex9 and *APOE* Ex2, identical to those used in the major isoforms of TOMM40 and APOE (see Discussion). In summary, this novel transcript translates into a protein that lacks the C-terminal 46 amino acid sequence of TOMM40, adds eight amino acids at the beginning of APOE, and retains the entire amino acid sequence of APOE (Figure 1B). Note that the same product was amplified using cDNA libraries from U937 cells, a human histiocytic lymphoma cell line, and HepG2 cells, a human hepatocellular carcinoma cell line (Figure 1C).

Based on the *TOMM40-APOE3*-expression plasmid containing HA-tag, we generated a *TOMM40-APOE4*-expression plasmid by introducing point mutations. We then transfected these plasmids into Neuro2a cells, a neuronal cell line, and confirmed the expression of these protein products using Western blotting (Figure 1D).

### 3.2. Intracellular Localization of APOE3, APOE4, TOMM40-APOE3 and TOMM40-APOE4

Next, to analyze the intracellular localization of TOMM40-APOE3 and TOMM40-APOE4, we performed immunostaining analyses using antibodies against HA, which recognize TOMM40-APOE-HA and Tom20 (a protein on the mitochondria), and found the co-localization of their signals (Figure 2A). On the other hand, the HA signals did not co-localize with KDEL (Figure 2B), an ER-localized protein, or GS28 (Figure 2C), a Golgi-localized protein, indicating that TOMM40-APOE is mainly localized to mitochondria, possibly owing to their N-terminal mitochondrial target sequences of TOMM40. We did not find any difference between TOMM40-APOE3 and TOMM40-APOE4 (Figure 2A–C). The same experiment was also performed for APOE3 and APOE4, but no co-localization with Tom20 (mitochondria) was observed. On the other hand, they co-localized well with GS28, possibly due to their secretion through the Golgi apparatus (Figure 2A–C).

### 3.3. Secretion of APOE3 and APOE4, but Not TOMM40-APOE3 and TOMM40-APOE4

Because APOE is an extracellularly secreted protein, we wondered whether TOMM40-APOE is also secreted or not. For this aim, we recovered culture media, collected HA-tagged proteins with immunoprecipitation using anti-HA antibody, and compared the secreted proteins with intracellular proteins using Western blotting. The results showed that APOE3 and APOE4 secreted substantial amounts of proteins extracellularly (Figure 2D), while secreted TOMM40-APOE3 and TOMM40-APOE4 were not observed (Figure 2E, red arrow). This is probably due to their mitochondrial localization, reflecting the original nature of TOMM40.

### 3.4. No Impact on Mitochondrial Membrane Potential by APOE3, APOE4, TOMM40-APOE3, and TOMM40-APOE4

Given that TOMM40-APOE3 and TOMM40-APOE4 localize on mitochondria, we considered their potential to affect mitochondrial function. Therefore, along with these genes, the GFP gene (a marker for transfected cells) was transfected, and 28 h later, MitoTracker Red (MTR), which fluoresces in a mitochondrial membrane potential-dependent manner, was administered, and the membrane potential was measured. Although four different genes were transfected, the number of cells expressing GFP and its fluorescence intensity were all about the same as in the control, suggesting that there was no difference in the efficiency of gene transfection (Figure 3A). When we analyzed the intensity of MTR fluorescence in GFP-positive cells, we did not find any effect of the expression of TOMM40-APOE3 or TOMM40-APOE4 (Figure 3B), suggesting that the expression of these proteins does not alter mitochondrial membrane potential. The same results were obtained when APOE3 and APOE4 were expressed (Figure 3A,B).

### 3.5. Induction of Cell Death by TOMM40-APOE3 and TOMM40-APOE4

Next, we analyzed cell death-inducing activity. For this aim, the piRFP670 gene and each TOMM40-APOE3 or TOMM40-APOE4 gene were co-transfected, and then propidium iodide (PI), a marker of cell death by staining plasma membrane disrupted cells, was administered time-coursely. Then, the population of PI-positive cells among the piRFP670-positive cells (transfected cells) was analyzed (Figure 3C). The results showed that cell death increased over time in TOMM40-APOE3- and TOMM40-APOE4-transfected cells, with approximately 30% of cells dying by 52 h (Figure 3C,D). Although APOE4 is associated with a higher risk of Alzheimer’s disease, no difference in cell death-inducing activity was observed between TOMM40-APOE3 and TOMM40-APOE4 (Figure 3C,D). Mild cell death was also observed with APOE3 and APOE4 expression (Figure 3C,D). Taken together, TOMM40-APOE3 and TOMM40-APOE4 are localized to mitochondria and are not exported outside the cell. Although they did not affect the mitochondrial membrane potential, they function to accelerate cell death through an unidentified mechanism. Within the scope of this analysis, no major qualitative differences between TOMM40-APOE3 and TOMM40-APOE4 were observed.

### 3.6. The Expression of TOMM40-APOE Transcripts in Human Brains

Finally, we confirmed the expression of read-through transcripts from the *TOMM40-APOE* locus in the hippocampus of human brains from eight AD and 10 non-AD subjects (Figure 4). Inter-individual variation in expression suggests underlying genetic variant effects as demonstrated in a previous study [4].

## 4. Discussion

Various read-through transcripts have been identified with the recent progress in genome analysis, and it has been suggested that these chimeric transcripts can be translated to generate fusion proteins that exhibit new subcellular localization or have special properties [17,18]. In fact, in this study, the fusion of the mitochondrial protein TOMM40 with the secreted protein APOE was found to generate a new mitochondrial protein without the properties of the latter.

TOMM40 is an essential protein for mitochondrial import and cell survival, as evidenced by the fact that mice deficient in TOMM40 are lethal in early embryonic stage [19]. If TOMM40-APOE acts antagonistically to TOMM40, the formation of mitochondrial membrane potential would be impaired, but in fact, there was no change in membrane potential in TOMM40-APOE-expressing cells, suggesting that TOMM40-APOE does not significantly inhibit the function of TOMM40 to a large extent. However, the possibility of a mild inhibitory effect cannot be denied, and further detailed mitochondrial protein import experiments will be important.

Because mitochondria are organelles that determine the life and death of cells, the effect on cell death was examined in this study [20,21]. As a result, it was observed that cells expressing TOMM40-APOE acted in a cytotoxic manner. Although further studies are needed on this mechanism, considering that TOMM40 is an outer membrane protein and does not affect the membrane potential formed inside and outside the inner membrane, it is possible that it enhances apoptotic membrane permeability on the mitochondrial outer membrane [22].

APOE has three isoforms, i.e., APOE2, E3, and E4, of which E4 is known as a risk gene for Alzheimer’s disease, multiple sclerosis, and other neurodegenerative disorders [8,23,24,25]. Therefore, we compared the chimeric protein with TOMM40 to see if there were any differences in its properties, but there seemed to be no significant differences as long as it functioned as a mitochondrial protein in the cell. In this analysis, TOMM40-APOE was expressed in differentiated Neuro2a cells to mimic neurons, considering its association with Alzheimer’s disease. However, since TOMM40-APOE was detected in cDNA derived from various cells (Figure 1), it is thought to be expressed to some extent in various organs in vivo. Therefore, it may exert stronger effects on mitochondria and whole cells, depending on the cell type, and more detailed studies, such as using transgenic mice, are needed.

The read-through transcripts utilize the canonical splice donor and acceptor sites at TOMM40 Ex9 and APOE Ex2. The underlying mechanism behind this alternative splicing event appears to involve non-strict recognition of the polyadenylation signal at the 3′-UTR (Ex10) of TOMM40, allowing for the extended transcription of pre-mRNA into APOE sequences. This so-called read-through transcript (pre-mRNA) tends to prefer the acceptor site at APOE Ex2 over TOMM40 Ex10, possibly due to structural changes in the pre-mRNA.

Regarding human brains, we did not find any significant difference in the expression of TOMM40-APOE transcripts between AD and non-AD subjects (Figure 4). The lack of significant results may be attributed to substantial inter-individual variation in transcript expression levels. This variation could require a larger sample size to detect differences with statistical significance. Additionally, variations in cellular fractions within brain samples may have contributed to this variability, as the read-through transcript is predominantly expressed in microglia. Furthermore, a previous study demonstrated that the AD risk allele of TOMM40 increases the expression of read-through transcripts [4], which could explain some of the inter-individual variation. Given the moderate genetic effect, we believe that additional samples, potentially hundreds, would be necessary to achieve statistical significance.

## Figures and Tables

**Figure 1 cells-13-00069-f001:**
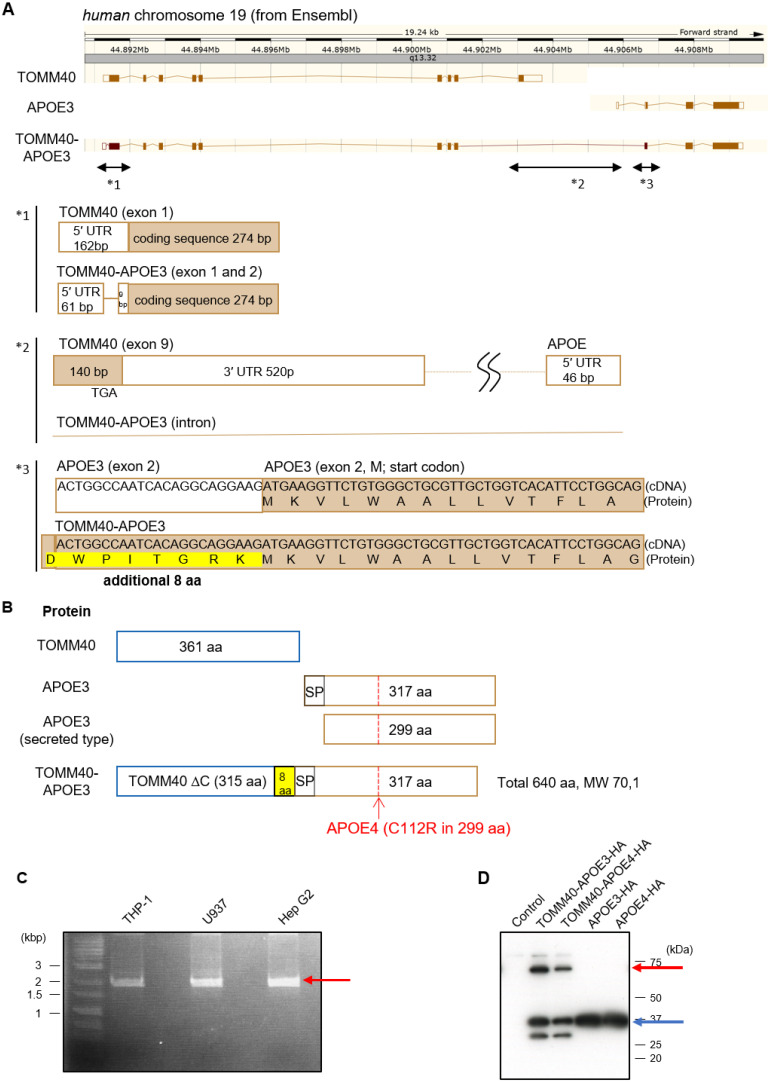
Purification of TOMM40-APOE products. (**A**) Schematic representation of the cDNA and amino acid sequence of *TOMM40-APOE*. Asterisks (*) 1–3 denoting the differing regions from *TOMM40* and *APOE* sequences: *1 indicates *TOMM40* exon1 vs. *TOMM40-APOE* exon 1 and 2; *2 indicates *TOMM40* exon9 (it includes the TGA stop codon) and the 5′ UTR of *APOE*. *TOMM40-APOE* mRNA skips exon9 and the 5′ UTR of APOE; and *3 indicates *TOMM40-APOE* cDNA has additional sequence before the *APOE* start ATG codon (23 bp 5′ UTR sequence in *APOE* exon2). (**B**) Schematic representation of APOE proteins. APOE3 has 18 aa signal peptide sequences (SP) in its N-terminal domain, and signal peptide sequences are deleted before secretion of APOE3. TOMM40-APOE has the TOMM40 domain (315 aa), 8 new peptides, and the APOE3 domain (full-length 317 aa). (**C**) Identification of TOMM40-APOE cDNA. PCR amplification was conducted using primer 1 (*TOMM40* N-terminal sequence) and primer 2 (*APOE* C-terminal sequence) with cDNA from THP-1, U937, and HepG2 cells as templates. The red arrow indicates the presence of bands corresponding to *TOMM40-APOE*. (**D**) Expression of TOMM40-APOE proteins. Neuro2a cells were transfected with vectors containing APOE3-HA, APOE4-HA, TOMM40-APOE3-HA, or TOMM40-APOE4-HA. After 28 h of transfection, cells were harvested and subjected to Western blot analyses. APOE3-HA and APOE4-HA proteins were applied at a 60-fold lower concentration than TOMM40-APOE3-HA and TOMM40-APOE4-HA due to the higher expression levels of APOE3 and APOE4. The red and blue arrows indicate bands corresponding to TOMM40-APOE3/4 and APOE3/4, respectively.

**Figure 2 cells-13-00069-f002:**
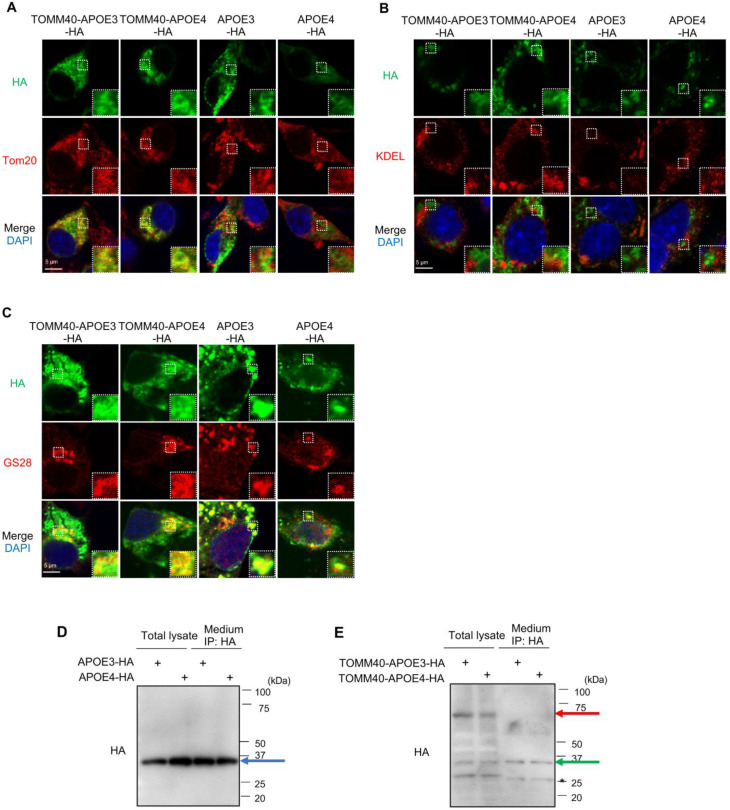
Intracellular localization of APOE3, APOE4, TOMM40-APOE3, and TOMM40-APOE4. (**A**–**C**) Localization of TOMM40-APOE. Neuro2a cells were transfected with the indicated genes. After 28 h of transfection, the cells were subjected to immunostaining with anti-HA antibody (shown in green) and specific organelle markers (indicated in red), including anti-Tom20 (**A**), KDEL (**B**), or GS28 (**C**). Subsequent observation was carried out using a confocal microscope. Yellow signals in the images denote areas of colocalization. (**D**,**E**) Secretion of APOE3 and APOE4, but not TOMM40-APOE3 and TOMM40-APOE4. Neuro2a cells were transfected with APOE3-HA and APOE4-HA (**D**) or TOMM40-APOE3-HA and TOMM40-APOE4-HA (**E**) for 4 h and then cultured with medium containing 2% FBS and 10 µM retinoic acid for neuronal differentiation. Then, 44 h after transfection, cells were harvested and lysed. Cell media were harvested, and immunoprecipitation was performed with an anti-HA antibody. Total lysates and immunoprecipitated proteins from the media were analyzed with Western blotting. Immunoprecipitates (lanes 3 and 4) were applied 2.5-fold more than total lysates (lanes 1 and 2). Blue and red arrows indicate the APOE3/4 band and TOMM40-APOE3/4 band, respectively. Note that TOMM40-APOE3/4 was detected in three bands (about 70, 35, and 30 kDa), and the proteins at 35 kDa (green arrow; the same size as APOE) were secreted. However, the proteins at 70 kDa (TOMM40-APOE3/4) were not secreted. * The position of the IgG light chain.

**Figure 3 cells-13-00069-f003:**
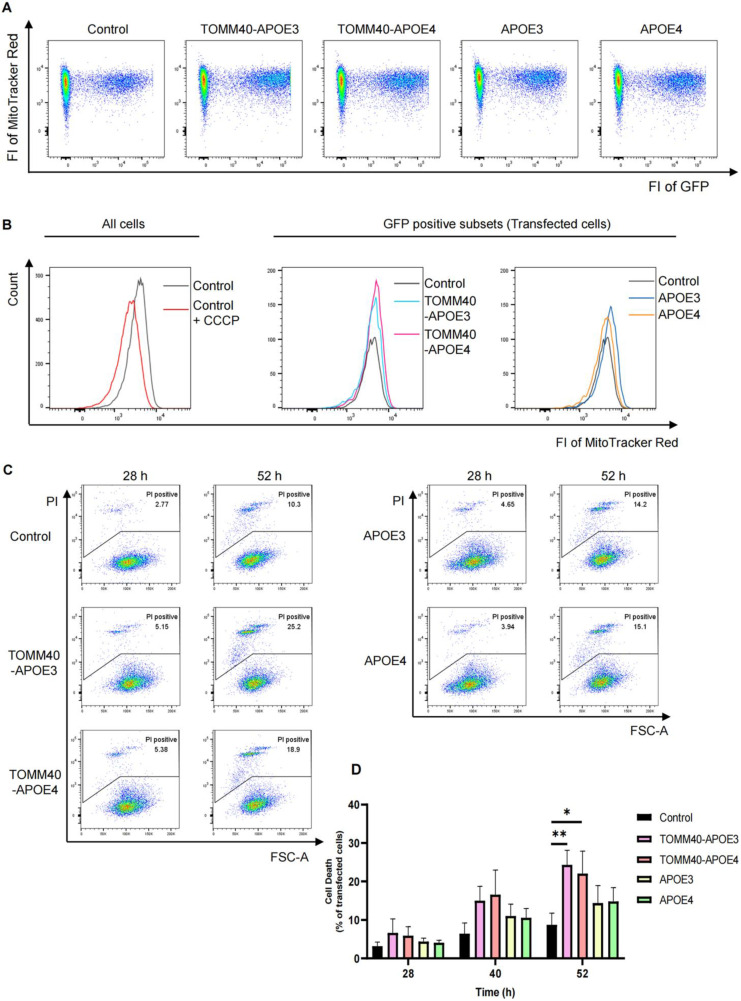
Induction of cell death without affecting mitochondrial membrane potential by TOMM40-APOE3 and TOMM40-APOE4. (**A**,**B**) No impact on mitochondrial membrane potential by APOE3, APOE4, TOMM40-APOE3, and TOMM40-APOE4. Neuro 2a cells were co-transfected with pmax-GFP vectors along with one of the following vectors: APOE3-HA, APOE4-HA, TOMM40-APOE3-HA, or TOMM40-APOE4-HA. After 28 h of transfection, MitoTracker Red staining was performed, allowing the assessment of MMP with flow cytometry. (**A**) The fluorescence of GFP (*x*-axis), representing transfected cells, and MitoTracker Red (*y*-axis), indicating mitochondrial membrane potential, were quantified. (**B**) Histogram displaying the intensity of MitoTracker Red fluorescence in transfected cells. CCCP was an uncoupler of oxidative phosphorylation. (**C**,**D**) Induction of cell death by TOMM40-APOE3 and TOMM40-APOE4. Neuro2a cells were co-transfected with piRFP670-N1 plasmid along with indicated plasmids. Then, 28 h, 40 h, and 52 h after transfection, cell death was assessed using propidium iodide (PI) staining and analyzed using flow cytometry. (**C**) Representative flow cytometric data at 28 h and 52 h. The *x*-axis indicates FSC-A and the *y*-axis indicates the fluorescence intensity of PI. The value of “PI positive” is the percentage of the number of dead transfected cells to the total number of transfected cells. (**D**) Bar graph depicting the percentage of deceased cells among transfected cells at 28 h, 40 h, and 52 h (*n* = 3). Asterisks (*) represent statistical significance: * *p* < 0.05 and ** *p* < 0.01.

**Figure 4 cells-13-00069-f004:**
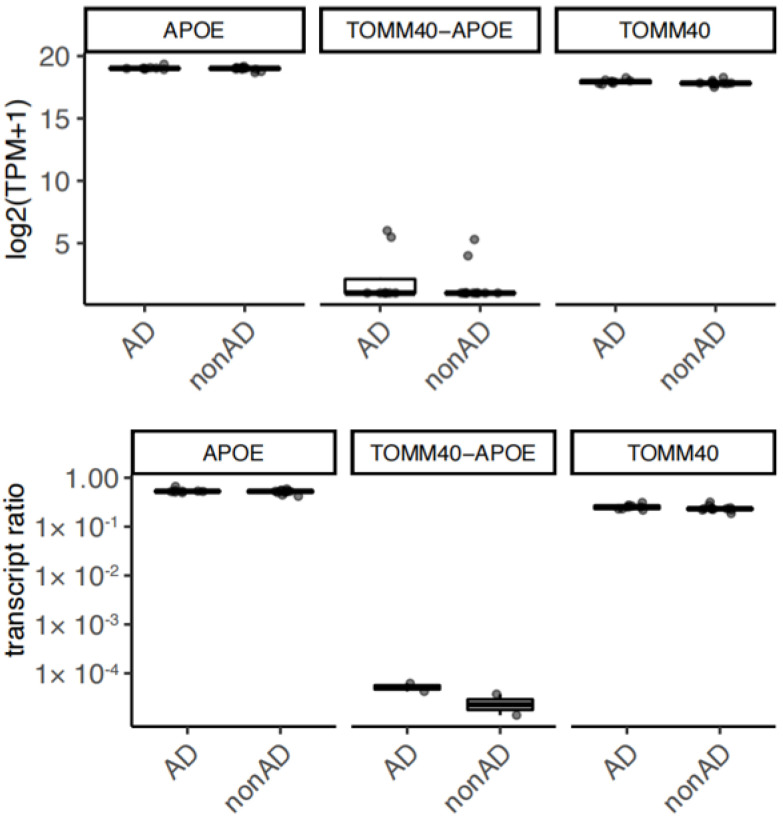
Expression of TOMM40-APOE transcripts in human brains. The read-through transcripts and the most major isoforms of APOE and TOMM40 (annotated as ENST00000485628.2 and 2a5cfe0e−e899−461f−ab42−706703f0aaf6, respectively, in a previous study: [4]) were quantified in the human autopsy brains from subjects with Alzheimer’s disease (AD) (*n* = 8) and subjects without AD (*n* = 10). The normalized expression level of transcripts (upper) and the transcript ratio over entire transcripts of TOMM40 and APOE (lower) are shown using box plots. TPM; transcripts per million.

## Data Availability

Data are contained within this article.

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
