# Peer review of "Uncovering the Localization and Function of a Novel Read-Through Transcript ‘TOMM40-APOE"

_cells, 2023, doi:10.3390/cells13010069_

Round 1

Reviewer 1 Report

Comments and Suggestions for Authors

The studies performed in this project were adequately conducted.

There are only minor points:

- The layout of Figure 4 should be changes in the way that the patients groups are shon on the X-axis while the log2(TPM+1) is diplayed on the Y-axis. Use Box-Plots

- Discuss in more detail the non-significant results obtained by analysing AD patients and controls. Try to give an explanation

Comments on the Quality of English Language

for example: line 13: mutations instead of mutation

Author Response

The studies performed in this project were adequately conducted.

There are only minor points:

Response:

We greatly appreciate this high evaluation of our manuscript.

Comment- The layout of Figure 4 should be changes in the way that the patients groups are shon on the X-axis while the log2(TPM+1) is diplayed on the Y-axis. Use Box-Plots

Response:

We appreciate this suggestion. We have replaced Figure 4 according to this recommendation. We have also explicitly mentioned in the figure legend that the presented graphs are Box-Plots to ensure readers understand the visualization method used.

Comment- Discuss in more detail the non-significant results obtained by analysing AD patients and controls. Try to give an explanation

Response:

We appreciate this comment. In accordance with this comment, we added the following discussion (page 12, lines 332-342). 

“Regarding human brains, we did not find any significant difference in expression of TOMM40-APOE transcripts between AD and non-AD subjects (Fig. 4). The lack of significant results may be attributed to substantial inter-individual variation in transcript expression levels. This variation could require a larger sample size to detect differences with statistical significance. Additionally, variations in cellular fractions within brain samples may have contributed to this variability, as the read-through transcript is predominantly expressed in microglia. Furthermore, a previous study demonstrated that the AD risk allele of TOMM40 increases the expression of read-through transcripts [4], which could explain some of the inter-individual variation. Given the moderate genetic effect, we believe that additional samples, potentially hundreds, would be necessary to achieve statistical significance.”

Comments on the Quality of English Language

for example: line 13: mutations instead of mutation

Response:

We appreciate this comment. We have corrected several typographical errors.

Reviewer 2 Report

Comments and Suggestions for Authors

Summary: This study aimed to address the biological consequences of a novel hybrid cDNA/protein that could impact the risk of Alzheimer’s disease. The originality and scientific soundness of the study is high.

General comments: Presentation of the data, interpretation and discussion of the results could be improved further.

Specific comments:

·       Reference #4 provides detailed information on the original finding of TOMM40-APOE hybrid cDNA. However, based on its citation number, it could not be retrieved. Providing a hyperlink (of bioRxiv PDF) will enhance accessibility. This reference should also be cited in Line 40.

·       A more detailed RNA splicing site connecting TOMM40 Ex9 to APOE Ex2 should be elaborated. For example, is this a canonical or non-canonical splicing site, and what is the potential explanation for this atypical splicing?

·       Detailed quantification data from PMB could provide insights into this hybrid cDNA/protein’s biological impact. Although expression levels of the hybrid cDNA are shown in Fig. 4, this data is not straightforward to grasp its actual quantity. Providing either fold changes or fractions of the hybrid cDNA, when compared to the original cDNAs of TOMM40 and APOE, will make this quantification much clearer.

·       Consider adding the following element to enrich the discussion. The hybrid TOMM40-APOE protein retains the original TOMM40 N-terminal sequence, which contains the mitochondrial localization peptide/signal. This might explain why this hybrid protein is mainly localized in mitochondria.

·       In Fig. 1, The “UTR” could be better labeled. For example: 5’-UTR vs. 3’-UTR.

Author Response

Summary: This study aimed to address the biological consequences of a novel hybrid cDNA/protein that could impact the risk of Alzheimer’s disease. The originality and scientific soundness of the study is high.

Response:

We greatly appreciate this high evaluation of our manuscript.

General comments: Presentation of the data, interpretation and discussion of the results could be improved further.

Response:

We have addressed all comments in full, as described below.

Specific comments:

Comment-· Reference #4 provides detailed information on the original finding of TOMM40-APOE hybrid cDNA. However, based on its citation number, it could not be retrieved. Providing a hyperlink (of bioRxiv PDF) will enhance accessibility. This reference should also be cited in Line 40.

Response:

We appreciate this comment. We added the hyperlink in References and cited Reference 4 in page 1 line 41 (line 40 in our previous manuscript).

Comment-·A more detailed RNA splicing site connecting TOMM40 Ex9 to APOE Ex2 should be elaborated. For example, is this a canonical or non-canonical splicing site, and what is the potential explanation for this atypical splicing?

Response:

Thank you very much for bringing this important issue to our attention. In accordance to this comment, we added the following sentence in RESULT section (page 4, lines160-163)

“The read-through transcripts utilize the canonical splice donor and acceptor sites at TOMM40 Ex9 and APOE Ex2, identical to those used in the major isoforms of TOMM40 and APOE.”

We have also added the following discussion (page 12, lines 325-331).  

“The read-through transcripts utilize the canonical splice donor and acceptor sites at TOMM40 Ex9 and APOE Ex2. The underlying mechanism behind this alternative splicing event appears to involve the non-strict recognition of the polyadenylation signal at the 3’-UTR (Ex10) of TOMM40, allowing for the extended transcription of pre-mRNA into APOE sequences. This so-called read-through transcript (pre-mRNA) tends to prefer the acceptor site at APOE Ex2 over TOMM40 Ex10, possibly due to structural changes in the pre-mRNA.”

Comment-·Detailed quantification data from PMB could provide insights into this hybrid cDNA/protein’s biological impact. Although expression levels of the hybrid cDNA are shown in Fig. 4, this data is not straightforward to grasp its actual quantity. Providing either fold changes or fractions of the hybrid cDNA, when compared to the original cDNAs of TOMM40 and APOE, will make this quantification much clearer.

Response:

We appreciate this helpful comment. To simplify matters, in our previous manuscript, we presented only the normalized expression levels (TPM) of the hybrid isoform, along with the major isoforms of TOMM40 and APOE. However, we agree that showing the fraction of transcripts would help, and have replaced Figure 4 by adding the transcript ratio of hybrid cDNA out of the entire TOMM40 and APOE isoforms.

Comment-·Consider adding the following element to enrich the discussion. The hybrid TOMM40-APOE protein retains the original TOMM40 N-terminal sequence, which contains the mitochondrial localization peptide/signal. This might explain why this hybrid protein is mainly localized in mitochondria.

Response:

We appreciate this kind comment. We described this point in the revised manuscript, as follows.

“TOMM40-APOE is mainly localized to mitochondria, possibly owing to their N-terminal mitochondrial target sequences of TOMM40.” (page 6, lines 197-198)

  • In Fig. 1, The “UTR” could be better labeled. For example: 5’-UTR vs. 3’-UTR.

Response:

We appreciate this comment. We made modification in Figure 1.